# Calculation and Allocation of Atmospheric Environment Governance Cost in the Yangtze River Economic Belt of China

**DOI:** 10.3390/ijerph20054281

**Published:** 2023-02-28

**Authors:** Jiekun Song, Zhicheng Liu, Rui Chen, Xueli Leng

**Affiliations:** School of Economics and Management, China University of Petroleum, Qingdao 266580, China

**Keywords:** atmospheric environment governance cost, modified Shapley value, modified fixed cost allocation model, fairness and efficiency, shadow price

## Abstract

Atmospheric environment governance requires necessary cost input. Only by accurately calculating regional atmospheric environment governance cost and scientifically allocating it within a region can the operability and realization of the coordinated governance of the regional environment be ensured. Firstly, based on the consideration of avoiding the technological regression of decision-making units, this paper constructs a sequential SBM-DEA efficiency measurement model and solves the shadow prices of various atmospheric environmental factors, that is, their unit governance costs. Secondly, combined with the emission reduction potential, the total regional atmospheric environment governance cost can be calculated. Thirdly, the Shapley value method is modified to calculate the contribution rate of each province to the whole region, and the equitable allocation scheme of the atmospheric environment governance cost is obtained. Finally, with the goal that the allocation scheme based on the fixed cost allocation DEA (FCA-DEA) model converges with the fair allocation scheme based on the modified Shapley value, a modified FCA-DEA model is constructed to achieve the efficiency and fairness of the allocation of atmospheric environment governance cost. The calculation and allocation of the atmospheric environmental governance cost in the Yangtze River Economic Belt in 2025 verify the feasibility and advantages of the models proposed in this paper.

## 1. Introduction

China’s economy has developed rapidly in recent years. Despite the impact of the COVID-19 pandemic, the GDP growth rate in 2020 was only 2.3%, but it recovered to 8.1% in 2021, reaching CNY 114,367 billion [1]. While vigorously developing the economy, the Chinese government has attached great importance to improving the ecological environment. According to the China Ecological Environment Bulletin 2021, 218 of the 339 cities surveyed met the standard of ambient air quality, which is 3.5 percentage points higher than that of 2020. At the same time, it should be noted that there are still 121 cities with air environment quality exceeding the standard, accounting for 35.7%. In addition, China has been the world’s largest CO_2_ emitter since 2007. According to the statistics of the International Energy Agency (IEA), China’s greenhouse gas emissions were 1286.1 billion tons of CO_2_ equivalent in 2021, which is an increase of 4.8% over the previous year. Therefore, China still needs to continue to strengthen atmospheric environment governance.

Atmospheric environmental emissions have spatial spillover effect [2,3], and the governance of them is comprehensive, systematic and regional, involving environmental governance planning, energy utilization, pollution prevention and control and other aspects [4,5,6]. In 2010, the Chinese government issued the Guidance on Promoting the Joint Prevention and Control of Air Pollution to Improve Regional Air Quality, in which Beijing–Tianjin–Hebei, Yangtze River Delta and Pearl River Delta were identified as key areas for joint prevention and control of air pollution. In the Law on the Prevention and Control of Air Pollution, it is proposed to strengthen the comprehensive prevention and control of air pollution, promote the joint prevention and control of regional air pollution, and implement coordinated control of atmospheric pollutants such as particulate matter, sulfur dioxide, nitrogen oxides and greenhouse gases. At the same time, it is clearly pointed out that provinces, autonomous regions and municipalities should strictly control or reduce the total amount of key atmospheric pollutants in their administrative areas. According to the Outline of the 14th Five-Year Plan (2021–2025) for National Economic and Social Development, China’s energy consumption per unit of GDP will decrease by 13.5%, CO_2_ emissions per unit of GDP will decrease by 18%, and the number of days with good air quality in cities at or above prefecture level will reach 87.5%. At the same time, China will strengthen the coordinated control of pollutants and regional coordinated governance, especially the implementation of major regional strategies, including promoting the coordinated prevention, control and governance of air pollution and the coordinated development of economy in Beijing–Tianjin–Hebei and promoting ecological and environmental protection and economic development in the Yangtze River Economic Belt. The Yangtze River Economic Belt is an important national strategic development area in China, with its population and GDP accounting for more than 40% of the country. Therefore, its atmospheric environment governance has a significant impact on China’s green economic development. Due to the fluidity and spatial spillover effect of atmospheric emissions, the nine provinces and two municipalities in the Yangtze River Economic Belt must establish a collaborative governance mechanism to jointly manage the atmospheric environment. The scientific calculation of regional atmospheric environment governance costs and reasonable allocation among provinces are the important basis of the coordinative governance mechanism.

Scholars have carried out fruitful studies on the calculation of the costs of atmospheric environmental governance. The commonly used calculation method is the unit governance cost method, which includes mainly the governance cost coefficient method and the marginal cost treatment method [7,8]. The governance cost coefficient is the ratio of actual cost to the amount of pollutant removed. This method is simple to calculate, but it is difficult to distinguish the governance costs of different emissions via this method. Furthermore, this method cannot reflect the urgency of environmental emission governance in different regions for their economic development. The marginal cost treatment method represented by the shadow price method has low data requirements and can reflect the impact of environmental emissions on the governance price from the perspective of environmental efficiency, which has been widely used by scholars [9,10]. The shadow price is interpreted as the opportunity cost of reducing an additional unit of bad output, which is often calculated based on efficiency measurement models. According to the classification of efficiency measurement models, shadow price methods include mainly the parametric method and the non-parametric method [11,12]. The parameter method is used mainly to calculate the environmental efficiency and shadow price through different production functions such as quadratic directional distance [13,14,15,16,17,18], non-radial distance [19], directional distance [20,21] and stochastic frontier [22]. The non-parametric method involves mainly data envelopment analysis (DEA), including SBM-DEA [23], SBM-DEA with unexpected output [24,25] and SBM-DEA with energy structure constraint [26].

Reasonable allocation of atmospheric environment governance costs is the basis for ensuring the efficiency of environmental governance, which is conducive to clarifying the responsibilities of various organizations and promoting the rational and effective use of funds. For the fixed cost allocation (FCA) problem, scholars mainly proposed DEA and co-operative game methods. The DEA method focuses on the efficiency of cost allocation, making the individual efficiency and the overall efficiency of allocation optimal [27,28,29,30]. In order to solve the non-uniqueness of the DEA distribution scheme, Li et al. [31] and Li et al. [32] further introduced the concepts of satisfaction and non-egoism, respectively. Co-operative game method focuses on the equity of allocation [33,34,35] and has been applied to the allocation of atmospheric environmental governance costs, mainly including the nucleolus method [36], the Nash negotiation model [37], the differential game model with fairness concern [38], and the Shapley value method [39,40,41].

A review of the literature reveals that there are still some deficiencies in the existing research that need to be further improved. (1) In the aspect of cost calculation, the shadow price method based on efficiency measure has become the mainstream for research, among which the non-parametric DEA method does not need to assume the form of production function in advance, so the research on this method is more extensive. However, in the existing research, it is difficult to eliminate the possible technical regression in the comparison with the historical years by analyzing only the input–output data of each DMU in the study year, resulting in the unobjective unit governance cost of each environmental element. (2) In the aspect of cost allocation, the DEA and Shapley methods are the two most studied methods, which reflect the efficiency and fairness of allocation, respectively. However, they cannot effectively take into account the fairness and efficiency, resulting in unscientific and inoperable final allocation results. (3) Few studies have been conducted on the calculation and allocation of atmospheric environment governance costs in the Yangtze River Economic Belt, but this demonstration study is of great importance for the implementation of China’s regional coordinated development strategy. To this end, this paper compensates for the above deficiencies, and its main contributions are as follows:(1)Considering the possible technological retrogression of each DMU, a sequential SBM-DEA efficiency measurement model is constructed, and the shadow price of each atmospheric environmental factor is calculated using duality theory. At the same time, the emission reduction potential of each factor is calculated based on environmental efficiency, and then the total cost of atmospheric environmental governance is calculated.(2)By combining the modified Shapley value model and the FCA-DEA model, an allocation model system of atmospheric environmental governance costs is established, which takes into account fairness and efficiency.(3)The above models are applied to calculate and allocate the atmospheric environment governance cost of the Yangtze River Economic Belt in 2025. The example verifies the feasibility of the model system and also provides decision support for the coordinated governance of the atmospheric environment in the Yangtze River Economic Belt.

The rest of this paper is organized as follows. Section 2 presents the research methods and explains the data sources. In Section 3, the above methods are used to obtain the calculation results. Section 4 summarizes the research conclusions of this paper.

## 2. Materials and Methods

### 2.1. Method

#### 2.1.1. Efficiency Measure Model Based on Sequential SBM-Undesirable

DEA model was first proposed by Charnes et al. [42]. Most traditional DEA models are radial and angular. Radial means that the input–output of the model expands or shrinks in proportion to achieve efficiency, and angular refers to whether the model is input-oriented or output-oriented. When the DMU is not effective, the efficiency value measured by the traditional DEA model is often overestimated. In order to overcome the radial and angular problems of the traditional DEA model, Tone [43] considered the slackness of the model’s input–output and proposed the SBM-Undesirable model. Although the SBM-Undesirable model solves the problems of radiality and angularity, it measures only the efficiency of a certain time and ignores the technology retrogression that may exist in the measurement process. Obviously, it is difficult to explain that the previously implemented technology is not feasible in the later stage. In order to solve this problem, this paper introduces the previous input–output data into the SBM-Undesirable model to build a sequential SBM-Undesirable model. The basic form is as follows:$$\rho_{0}^{*}=\min\frac{1-\frac{1}{K}\sum_{k=1}^{K}\frac{s_{k}^{x}}{x_{k0}}}{1+\frac{1}{M+I}\left(\sum_{m=1}^{M}\frac{s_{m}^{y}}{y_{m0}}+\sum_{i=1}^{I}\frac{s_{i}^{b}}{b_{i0}}\right)}\;$$
s.t.
(1)$$x_{k0}^{T}=\sum_{t=1}^{T}\sum_{j=1}^{n}z_{j}^{t}x_{kj}^{t}+s_{k}^{x},\;k=1,\ldots ,K\;y_{m0}^{T}=\sum_{t=1}^{T}\sum_{j=1}^{n}z_{j}^{t}y_{mj}^{t}-s_{m}^{y},\;m=1,\ldots ,M\;b_{i0}^{T}=\sum_{t=1}^{T}\sum_{j=1}^{n}z_{j}^{t}b_{ij}^{t}+s_{i}^{b},\;i=1,\ldots ,I\;s_{k}^{x}\geq 0,s_{m}^{y}\geq 0,s_{i}^{b}\geq 0,z_{j}^{t}\geq 0$$
where period *t* = 1 … *T*; $x_{j}^{t}$, $y_{j}^{t}$ and $b_{j}^{t}$ are the input, desired output and non-desired output of DMU*_j_* in the period *t*, respectively. $z_{j}^{t}$ is the decision variable in the period *t* of DMU*_j_*. According to the environmental efficiency obtained from the sequential SBM-Undesirable model, the abatement potential of pollution $APP_{i,j}$ of the *i*th pollutant of the DMU*_j_* can be calculated by comparing the actual pollution emissions with the target pollution emissions on the frontier of DMU*_j_*, and the calculation formula is:(2)$$APP_{i,j}=\frac{s_{i,j}}{AP_{i,j}}$$
where $s_{i,j}$ and $AP_{i,j}$ are the redundancy and actual emission of the *i*th undesirable output of DMU*_j_*, respectively. The larger the $APP_{i,j}$ is, the greater the emission of the *i*th pollutant of DMU*_j_* is, and the greater the emission reduction potential of the *i*th pollutant of DMU*_j_* is.

The model (1) is a nonlinear programming model. Through Charnes–Cooper transform:$$q=\frac{1}{1+\frac{1}{M+I}\left(\sum_{m=1}^{M}\frac{s_{m}^{y}}{y_{m0}}+\sum_{i=1}^{I}\frac{s_{i}^{b}}{b_{i0}}\right)},\;S_{k}^{x}=qs_{k}^{x},\;S_{m}^{y}=qs_{m}^{y},\;S_{i}^{b}=qs_{i}^{b},\;\eta_{i}^{b}=qz_{j}^{t}$$
it can be transformed into a linear programming form:$$\rho_{0}^{*}=Min\left[q-\frac{1}{k}\left(\overset{K}{\underset{k=1}{\sum }}\frac{S_{k}^{x}}{x_{k0}}\right)\right]$$
s.t.
(3)$$1=q+\frac{1}{M+I}\left(\sum_{m=1}^{M}\frac{S_{m}^{y}}{y_{m0}}+\sum_{i=1}^{I}\frac{S_{i}^{b}}{b_{i0}}\right)\;qx_{k0}^{T}=\overset{T}{\underset{t=1}{\sum }}\overset{n}{\underset{j=1}{\sum }}\eta_{j}^{t}x_{kj}^{t}+S_{k}^{x},\;k=1,\ldots ,K\;qy_{m0}^{T}=\sum_{t=1}^{T}\sum_{j=1}^{n}\eta_{j}^{t}y_{mj}^{t}-S_{m}^{y},\;m=1,\ldots ,M\;qb_{i0}^{T}=\sum_{t=1}^{T}\sum_{j=1}^{n}\eta_{j}^{t}b_{ij}^{t}+S_{i}^{b},\;i=1,\ldots ,I\;S_{k}^{x}\geq 0,S_{m}^{y}\geq 0,S_{i}^{b}\geq 0,\eta_{j}^{t}\geq 0,q\geq 0$$

#### 2.1.2. Calculation Model of Environmental Governance Cost Based on Shadow Price

To solve the shadow price of each environmental element, according to the dual theory, the dual problem of model (3) is:$$\xi^{*}=\max\xi$$
s.t.
(4)$$\sum_{k=1}^{K}x_{k0}^{T}u_{k}-\sum_{m=1}^{M}y_{m0}^{T}v_{m}+\sum_{i=1}^{I}b_{i0}^{T}v_{bi}+\xi \leq 1\;\sum_{m=1}^{M}y_{mj}^{t}v_{m}-\sum_{k=1}^{K}x_{kj}^{t}u_{k}-\sum_{i=1}^{I}b_{ij}^{t}v_{bi}\leq 0,\;j=1,\ldots ,n;\;t=1,\ldots ,T\;u_{k}\geq \frac{1}{Kx_{k0}^{T}},\;k=1,2,\ldots ,K\;v_{m}\geq \frac{\xi }{\left(M+I\right)y_{m0}^{T}},\;m=1,2,\ldots ,M\;v_{bi}\geq \frac{\xi }{\left(M+I\right)b_{i0}^{T}},\;i=1,2,\ldots ,I\;u_{k}\geq 0,v_{m}\geq 0,v_{bi}\geq 0\;\xi \;\mathrm{is}\;\mathrm{unconstrained}$$
where $u_{k}$, $v_{m}$ and $v_{bi}$ are the decision variables of the dual problem corresponding to the constraints of the original problem, which represent the virtual prices of input, desirable output and undesirable output, respectively.

In order to solve the problem easily, the model (4) can be transformed equivalently as follows:$$max\;\overset{M}{\underset{m=1}{\sum }}y_{m0}^{T}v_{m}-\overset{K}{\underset{k=1}{\sum }}x_{k0}^{T}u_{k}-\overset{I}{\underset{i=1}{\sum }}b_{i0}^{T}v_{bi}$$
s.t.
(5)$$\sum_{m=1}^{M}y_{mj}^{t}v_{m}-\sum_{k=1}^{K}x_{kj}^{t}u_{k}-\sum_{i=1}^{I}b_{ij}^{t}v_{bi}\leq 0,\;j=1,\ldots ,n;\;t=1,\ldots ,T\;u_{k}\geq \frac{1}{Kx_{k0}^{T}},\;k=1,\;\ldots ,K\;v_{m}\geq \frac{1+\sum_{m=1}^{M}y_{m0}^{T}v_{m}-\sum_{k=1}^{K}x_{k0}^{T}u_{k}-\sum_{i=1}^{I}b_{i0}^{T}v_{bi}}{\left(M+I\right)y_{m0}^{T}},\;m=1,\ldots ,M\;v_{bi}\geq \frac{1+\sum_{m=1}^{M}y_{m0}^{T}v_{m}-\sum_{k=1}^{K}x_{k0}^{T}u_{k}-\sum_{i=1}^{I}b_{i0}^{T}v_{bi}}{\left(M+I\right)b_{i0}^{T}},\;i=1,\ldots ,I\;u_{k}\geq 0,v_{m}\geq 0,v_{bi}\geq 0$$

The objective function of the model (5) can be viewed as virtual profit. It is assumed that the shadow prices of input, desirable output and undesirable output are *P_x_*, *P_y_* and *P_b_*, respectively. According to the relationship between the shadow prices of desirable output and undesirable output, assume that the shadow price of GDP, the desirable output, *P_y_* = 1, then the shadow price of undesirable output *P_bi_* can be expressed as:(6)$$P_{bi}=P_{y}\frac{v_{bi}}{v_{m}}=\frac{v_{bi}}{v_{m}}$$

Formula (6) reflects the opportunity cost needed to reduce an additional unit of undesirable output, namely the marginal emission reduction cost. This paper takes the shadow price of undesirable output as its unit governance cost to carry out the following research.

In order to determine the total regional atmospheric environment governance cost, it is necessary to ensure that the environmental efficiency of all provinces and cities in the region is effective. Therefore, assuming that the redundancy of all undesirable outputs of each DMU in the region is effectively controlled, that is, all DMUs reach the production frontier, the required governance cost in this case is taken as the atmospheric environment governance cost, which is expressed as:(7)$$C=\overset{n}{\underset{j=1}{\sum }}C_{j}=\overset{n}{\underset{j=1}{\sum }}\overset{I}{\underset{i=1}{\sum }}P_{bi}s_{i}^{b}\;$$
where *C_j_* refers to the atmospheric environment governance cost of DMU*_j_*, $P_{bi}$ and $s_{i}^{b}$ are the unit governance cost (that is, shadow price) and the governance amount of the *i*th undesirable output, respectively.

#### 2.1.3. Equitable Allocation of Regional Atmospheric Environment Governance Cost

In order to ensure the fairness and rationality of cost allocation, the efficiency of all possible alliances formed by all DMUs in the region is analyzed based on co-operative game theory. If there are *n* DMUs in the region, 2*^n^*−1 alliances will be formed. The input–output data of all DMUs in each alliance are brought into the sequential SBM model to obtain the environmental efficiency value of each DMU under 2*^n^*−1 alliances. Then, based on the modified Shapley value, the individual contribution of each DMU is calculated, and the cost is allocated fairly. For an input–output production system with *n* players, the modified Shapley value $\varphi \left(v\right)=(\varphi_{1}\left(v\right),\varphi_{2}\left(v\right),\ldots ,\varphi_{n}\left(v\right))$ is defined as:(8)$$\varphi_{i}\left(v\right)=\sum_{S\subset N}w\left|S\right|\frac{\left[v\left(S\right)/v\left(S/i\right)\right]}{v_{i}\left(S\right)/v_{i}\left(\left\{i\right\}\right)},\;i=1,2,\ldots ,n$$
where v(S) is the characteristic function of alliance *S*:(9)$$v\left(S\right)=\sum_{i\in S}v_{i}\left(S\right)=\sum_{i\in S}\rho_{i}$$

Here, $v_{i}\left(S\right)$ represents the characteristic function of the player *i* in the alliance *S*, and its value is equal to the environmental efficiency value of DMU*_i_* in *S*.

w|S| is the statistical probability when the player *i* appears in the |S|-th position:(10)$$w\left|S\right|=\frac{\left|S-1\right|!\left(n-\left|S\right|\right)!}{n!}$$

Here, |S| represents the number of players included in the alliance *S*.

v(S)/v(S/i) reflects the impact of player *i* on the efficiency of other players before and after joining the alliance *S*. The larger the value of v(S)/v(S/i), the higher the effectiveness of player *i* to other players in the alliance *S*, and the greater the contribution to the alliance *S*.

$v_{i}\left(S\right)/v_{i}\left(\left\{i\right\}\right)$ reflects the change of the efficiency value of player *i* before and after joining the alliance *S*. Because the feasible region of the sequential SBM model becomes larger after the player *i* joins the alliance, the value of the objective function is less than or equal to the value before the player *i* joins the alliance. Therefore, the larger the value of $v_{i}\left(S\right)/v_{i}\left(\left\{i\right\}\right)$, the more effective the player *i* is. It can be seen that Formula (8) represents the contribution of player *i* to alliance *S*.

Because $v_{i}\left(\left\{i\right\}\right)$ includes only the player *i*, its optimal efficiency value is 1, and Formula (8) can be expressed as:(11)$$\varphi_{i}\left(v\right)=\sum_{S\subset N}w\left|S\right|\frac{\left[v\left(S\right)/v\left(S/i\right)\right]}{v_{i}\left(S\right)}$$

The modified Shapley value reflects the relative contribution of each player to the whole system. Convert the relative contribution degree into the contribution rate of the players in the system, and then the cost allocation of each player in the system can be carried out according to the following proportion:(12)$$V=\left(\frac{\varphi_{1}\left(v\right)}{\sum_{i=1}^{n}\varphi_{i}\left(v\right)},\ldots ,\frac{\varphi_{n}\left(v\right)}{\sum_{i=1}^{n}\varphi_{i}\left(v\right)}\right)$$

#### 2.1.4. Allocation Model and Solution of Regional Atmospheric Environment Governance Cost Based on Modified FCA-DEA Model

The FCA-DEA model allocates the fixed cost as a new input among the DMUs. It assumes that the cost to be allocated has strong disposability and believes that each DMU has individual rationality and collective rationality, that is, each DMU pursues the maximization of its own efficiency and the overall efficiency. Taking the fixed cost and undesirable outputs as inputs and aiming to maximize the average efficiency of all DMUs [44], the FCA-DEA model can be constructed as follows [26]:$$max\;E=\frac{1}{n}\overset{n}{\underset{j=1}{\sum }}e_{j}$$
s.t.
(13)$$\frac{\sum_{m=1}^{M}\alpha_{m}y_{mj}}{f_{j}+\sum_{k=1}^{K}\beta_{k}x_{kj}+\sum_{i=1}^{I}\lambda_{i}u_{ij}}=e_{j},\;j=1,2,\ldots ,n\;0\leq e_{j}\leq 1\;\sum_{j=1}^{n}f_{j}=F\;\alpha_{m}\geq 0,\;m=1,2,\ldots ,M\;\beta_{k}\geq 0,\;k=1,2,\ldots ,K\;\lambda_{i}\geq 0,\;i=1,2,\ldots ,I$$
where $e_{j}$ is the input–output efficiency of DMU*_j_*, and $f_{j}$ is the cost allocated to DMU*_j_* under the premise of overall efficiency maximization. The sum of costs apportioned by all DMU must be equal to the total cost to be apportioned, and $\beta_{k}$, $\alpha_{m}$ and $\lambda_{i}$ are the weight vectors of input, desirable output and undesirable output, respectively.

The optimal objective function *E*^*^ = *max E* is included in the model (13) as a constraint. Under the condition that the overall optimal efficiency remains unchanged, the model is constructed with the goal of minimizing and maximizing the cost allocated to DMU*_j_* as follows:$$max\;f_{j}\left(min\;f_{j}\right)$$
s.t.
(14)$$\frac{\sum_{m=1}^{M}\alpha_{m}y_{mj}}{f_{j}+\sum_{k=1}^{K}\beta_{k}x_{kj}+\sum_{i=1}^{I}\lambda_{i}u_{ij}}=e_{j},\;j=1,2,\ldots ,n\;0\leq e_{j}\leq 1\;\frac{1}{n}\sum_{j=1}^{n}e_{j}\geq E^{*}\;\sum_{j=1}^{n}f_{j}=F\;\alpha_{m}\geq 0,\;m=1,2,\ldots ,M\;\beta_{k}\geq 0,\;k=1,2,\ldots ,K\;\lambda_{i}\geq 0,\;i=1,2,\ldots ,I\;f_{j}\geq 0,\;j=1,2,\ldots ,n$$

The values of the two objective functions $max\;f_{j}$ and $min\;f_{j}$ are respectively denoted as $U_{j}$ and $L_{j}.$ For DMU*_j_*, the allocated cost $f_{j}$ satisfies $L_{j}\leq f_{j}\leq U_{j}$. Theoretically, when $f_{j}$ takes the value in $\left[L_{j},U_{j}\right]$, the overall efficiency is optimal.

The model (14) gives the value range of the allocated cost of DMU*_j_*. From the perspective of a rational person, each DMU tends to obtain the lowest cost on the premise of ensuring the optimal overall efficiency, the result of which may be that the cost to be allocated cannot be fully allocated. In addition, the model considers only efficiency and ignores the fairness of allocation. To this end, this paper integrates the above fair scheme with the FCA-DEA model and takes the efficient allocation scheme converging with the fair scheme as the objective function to modify the FCA-DEA model. The model is as follows:$$min\overset{n}{\underset{j=1}{\sum }}\left|f_{j}-f_{j}^{0}\right|$$
s.t.
(15)$$\frac{\sum_{m=1}^{M}\alpha_{m}y_{mj}}{f_{j}+\sum_{k=1}^{K}\beta_{k}x_{kj}+\sum_{i=1}^{I}\lambda_{i}u_{ij}}=e_{j},\;j=1,2,\ldots ,n\;L_{j}\leq f_{j}\leq U_{j},\;j=1,2,\ldots ,n\;0\leq e_{j}\leq 1\;\frac{1}{n}\sum_{j=1}^{n}e_{j}\geq E^{*}\;\sum_{j=1}^{n}f_{j}=F\;\alpha_{m}\geq 0,\;m=1,2,\ldots ,M\;\beta_{k}\geq 0,\;k=1,2,\ldots ,K\;\lambda_{i}\geq 0,\;i=1,2,\ldots ,I\;f_{j}\geq 0,\;j=1,2,\ldots ,n$$
where $f_{j}^{0}$ is the cost allocated under the principle of fairness.

The objective function of model (13) contains absolute value, which can be converted for ease of solution. Let $a_{j}=\left(\left|f_{j}-f_{j}^{0}\right|+f_{j}-f_{j}^{0}\right)/2$ and $b_{j}=\left(\left|f_{j}-f_{j}^{0}\right|-f_{j}+f_{j}^{0}\right)/2$; then, the model (15) becomes
$$min\overset{n}{\underset{j=1}{\sum }}\left(a_{j}-b_{j}\right)$$
s.t.
(16)$$\overset{M}{\underset{m=1}{\sum }}\alpha_{m}y_{mj}-\overset{K}{\underset{k=1}{\sum }}\beta_{k}x_{kj}-\overset{I}{\underset{i=1}{\sum }}\lambda_{i}u_{ij}-a_{j}+b_{j}=f_{j}^{0}\;\sum_{j=1}^{n}\left(a_{j}-b_{j}\right)=0\;L_{j}\leq a_{j}-b_{j}+f_{j}^{0}\leq U_{j},\;j=1,2,\ldots ,n\;\;\alpha_{m}\geq 0,\;m=1,2,\ldots ,M\;\beta_{k}\geq 0,\;k=1,2,\ldots ,K\;\lambda_{i}\geq 0,\;i=1,2,\ldots ,I\;a_{j},b_{j}\geq 0,\;j=1,2,\ldots ,n$$

Let the optimal solutions be $a_{j}^{*}$, $b_{j}^{*}$, $\alpha_{m}^{*}$, $\beta_{k}^{*}$ and $\lambda_{i}^{*}$; then, the optimal cost allocation scheme is as follows:(17)$$f_{j}^{*}=a_{j}^{*}-b_{j}^{*}+f_{j}^{0},\;j=1,2,\ldots ,n$$

### 2.2. Data Sources and Processing

Referring to the studies of scholars [25,45,46,47,48], this paper selects population, capital stock and energy consumption as inputs, GDP as desirable output, and CO_2_ emission, NO_X_ emission and PM_2.5_ concentration as undesirable outputs to construct an environmental efficiency measurement model.

(1)Data from 2013 to 2020

Yangtze River Economic Belt covers nine provinces and two municipalities, (for the convenience of marking, the following are called provinces), namely Shanghai, Jiangsu, Zhejiang, Anhui, Jiangxi, Hubei, Hunan, Chongqing, Sichuan, Guizhou and Yunnan. The data of population, capital stock and GDP of 11 provinces in the Yangtze River Economic Belt are from the China Statistical Yearbook 2014–2021. GDP data are converted into the values at the constant price in 2020 according to the indices of GDP. The capital stock data are calculated by using the perpetual inventory method developed by OECD [49], in which the value of total fixed assets and fixed asset investment price index are derived from the China Statistical Yearbook. Because China has no longer counted the total fixed assets since 2017, this paper calculates the capital stock of 11 provinces from 2013 to 2017 at constant prices in 2015. Through linear fitting of the capital stock data of each province from 2013 to 2017, it is found that their R^2^ values of the fitting equations are higher than 0.9. Therefore, the capital stock data of 11 provinces from 2018 to 2020 are predicted by the fitting equations. The CO_2_ emission data are calculated via the IPCC emission factor method of energy consumption, in which the energy consumption data are from the China Energy Statistical Yearbook. The NO_x_ emission data are from the China Statistical Yearbook on Environment 2014–2021. The PM_2.5_ concentration data of each province refer to the relevant research of Washington University in St. Louis [50], which is expressed by the annual average emission concentration.

(2)Data from 2021 to 2025

Based on the GDP in 2020 and the target growth rate set in the “Fourteenth Five Year Plan” of each province, this paper calculates the GDP of each province from 2021 to 2025 at constant prices in 2020. Based on the average population growth rate of each province during the 13th Five Year Plan period and assuming that the population continues to grow or decline at this rate during the 14th Five Year Plan period, the population of each province from 2021 to 2025 can be obtained. Among the 11 provinces, the population growth rates of Anhui, Jiangxi, Hubei, Hunan and Yunnan are negative, and those of the other six provinces are positive. The capital stock data from 2021 to 2025 are obtained in the same way as those from 2018 to 2020 by using the prediction method. Based on the energy consumption intensity values of all provinces in 2020 and the decreasing targets of energy consumption intensity set in their “Fourteenth Five Year Plan”, the energy consumption intensity values of 11 provinces in 2025 are obtained. The energy conservation targets are allocated to each year according to the annual energy consumption intensity decreasing rate, and the energy consumption of each province from 2021 to 2025 can be calculated by multiplying the GDP forecast value of the corresponding year. The data of the CO_2_ emissions, the NO_x_ emissions and the PM_2.5_ concentration of each province from 2021–2025 are obtained in the same way as the energy consumption is obtained.

Table 1 and Table 2 are descriptive statistics of variables from 2021 to 2025 and forecast values of input and output indicators in 2025, respectively. Among them, the indicator of PM_2.5_ concentration is the annual average PM_2.5_ emission concentration of each province. According to the “WHO Air quality guidelines for particulate matter, ozone, nitrogen dioxide and sulfur dioxide”, the final target of annual average PM_2.5_ emission concentration is less than 10 µg/m^3^. It can be seen from the data in Table 1 that there is still a long way to go for PM_2.5_ emission reduction in the 11 provinces.

## 3. Results

### 3.1. Atmospheric Environmental Governance Cost in Yangtze River Economic Belt

#### 3.1.1. The Shadow Price of CO_2_ Emissions

Figure 1 shows the shadow price of CO_2_ emissions in 11 provinces of the Yangtze River Economic Belt from 2013 to 2025. It can be seen that the shadow price of CO_2_ emissions varies greatly among provinces in the Yangtze River Economic Belt. Although the shadow price of each province fluctuates, their overall trend is upward. This shows that with the increase in emission reduction efforts in each province, the marginal cost of reducing emissions is also rising, such that the pressure of carbon emission reduction is increasing.

By comparing the average CO_2_ emission shadow price and the average CO_2_ emission intensity of 11 provinces from 2013 to 2025 (see Table 3), it can be found that there is a correlation between shadow price and emission intensity. Generally speaking, the provinces with lower CO_2_ emission intensity have higher shadow price, or higher emission reduction costs, and the more difficult it is to continue to reduce emissions. The average shadow price of CO_2_ emissions in the Yangtze River Economic Belt is CNY 1432.65/ton, and the CO_2_ emission intensity is CNY 1.148 tons/10^4^. Zhejiang, Shanghai, Sichuan, Chongqing and Hunan rank as the top five in terms of CO_2_ shadow price, and their emission intensity values are also relatively low. The CO_2_ emission intensity of Zhejiang, Shanghai, Sichuan, Chongqing and Hunan provinces is lower than CNY 1.1 tons/10^4^, indicating that these provinces have a higher level of economic development, more advanced production technology and higher energy efficiency than other provinces, and the economic costs of further reducing CO_2_ emissions are also greater. Guizhou, Anhui, Jiangxi, Yunnan and Jiangsu rank as the last five provinces in terms of CO_2_ shadow price, which is lower than CNY 1200/ton for all regions, and their emission intensity values are also relatively large.

#### 3.1.2. The Shadow Price of NO_X_ Emissions

Figure 2 shows the change trend in the shadow price of NO_X_ emissions in each province from 2013 to 2025. It can be seen that the shadow price of NO_X_ emissions in different provinces varies greatly. The average shadow prices of NO_X_ emissions in Shanghai, Jiangsu and Chongqing rank as the top three, with CNY 313.64 ten thousand/ton, CNY 216.75 ten thousand/ton and CNY 196.98 ten thousand/ton, respectively, whereas the shadow price in Sichuan is the lowest, with only CNY 27.97 ten thousand/ton. The average shadow price of the NO_x_ emissions of 11 provinces in the Yangtze River Economic Belt is CNY 146.50 ten thousand/ton. Except for the fact that the shadow price of Sichuan in 2025 is lower than that in 2013, the marginal emission reduction costs of other provinces increase more than twice during the study period. The NO_X_ emission reduction cost in each province is generally on the rise, which indicates that with the continuous implementation of energy conservation and emission reduction policies, it is increasingly difficult to reduce NO_X_ emissions in the Yangtze River Economic Belt.

#### 3.1.3. The Shadow Price of PM_2.5_ Emissions

During the study period, the shadow price of PM_2.5_ emissions in each province of the Yangtze River Economic Belt is much higher than the shadow price of CO_2_ emissions and NO_X_ emissions. The shadow price of 11 provinces from 2013 to 2025 ranges from CNY 354.58 billion/(μg/m^3^) to CNY 27,683.57 billion/(μg/m^3^), and their average shadow price is CNY 3529.76 billion/(μg/m^3^). Figure 3 shows the change trend in the shadow price of PM_2.5_ emissions in each province from 2013 to 2025. It can be seen that the emission reduction cost in each province shows an upward trend, indicating that the Yangtze River Economic Belt has achieved significant results in terms of PM_2.5_ governance. The shadow price of PM_2.5_ emissions in Sichuan is far higher than that in other provinces. The reason is that during the period of 2013–2025, the average PM_2.5_ concentration of Sichuan is only 20.14 μg/m^3^, which is far lower than that of other provinces. The average PM_2.5_ emission reduction costs in Sichuan, Jiangsu and Zhejiang rank as the top three, with CNY 17,296.39 billion/(μg/m^3^), CNY 3773.68 billion/(μg/m^3^) and CNY 3765.91 billion/(μg/m^3^), respectively. Guizhou has the lowest average emission reduction cost at CNY 949.08 billion/(μg/m^3^).

#### 3.1.4. Emission Reduction Potential and Total Governance Cost of Atmospheric Environment in 2025

Table 4 shows the environmental efficiency and emission reduction potential of CO_2_, NO_X_ and PM_2.5_ in 11 provinces in 2025. The environmental efficiency values of 11 provinces range from 0.41 to 1, with an average value of 0.68, which is relatively low on the whole. The frontier areas of environmental efficiency are Shanghai, Jiangsu and Zhejiang on the east coast and Sichuan in the Chengdu Chongqing Economic Circle on the upper reaches of the Yangtze River. Their environmental efficiency values are 1, and their emission reduction potential is 0, that is, they achieve an optimal allocation of inputs and outputs without any emission redundancy in the relative assessment of 11 provinces. The environmental efficiency values of the other seven provinces are lower than the average value. Among them, the environmental efficiency values of Guizhou and Yunnan are lower than 0.5, both of which are located in the west of the Yangtze River Economic Belt. From the perspective of spatial distribution, except for Sichuan, the overall environmental efficiency is higher in the east and lower in the west. In addition, Jiangsu has high environmental efficiency, but its CO_2_ and NO_X_ emissions and PM_2.5_ concentration are also high. The two seem contradictory, but they are not. Although Jiangsu has developed in terms of science and technology and high environmental efficiency, it has a large population density. In order to meet the production and living needs and the sustained economic growth, it needs to consume a huge amount of energy. Therefore, the emissions of CO_2_ and NO_X_ and the density of PM_2.5_ are large, reducing the environmental quality of the province.

The average emission reduction potential of CO_2_, NO_X_ and PM_2.5_ in the Yangtze River Economic Belt in 2025 will be 22.09%, 33.99% and 8.53%, respectively. Guizhou has the largest CO_2_ emission reduction potential, which is 63.12%, nearly two times higher than the average emission reduction potential. Its NO_x_ emission reduction potential is 70.57%, which is also far higher than the average. In addition, Anhui, Jiangxi and Yunnan also have high CO_2_ emission reduction potential, which is 49.3%, 45.3% and 32.89%, respectively. Guizhou, Anhui, Jiangxi and Yunnan are the major provinces that can reduce CO_2_ emissions. In 2025, the CO_2_ emissions that can be reduced will account for 23.77%, 28.63%, 16.58% and 10.71% of the emissions of the Yangtze River Economic Belt, respectively. They are the key monitoring areas for CO_2_ emissions. As for NO_X_, except for Guizhou, the regions of Yunnan, Anhui, Hubei and Jiangxi also have huge emission reduction potential, which is 65.38%, 59.53%, 61.02% and 58.85%, respectively. Guizhou, Yunnan, Anhui, Hubei and Jiangxi are the major provinces that can reduce NO_x_ emissions, accounting for 15.28%, 17.74%, 19.87%, 23.93% and 13.31% of the Yangtze River Economic Belt, respectively. As for PM_2.5_, only Anhui, Jiangxi, Chongqing and Guizhou have emission reduction potential, which will be 11.96%, 15.28%, 34.31% and 32.25%, respectively. Chongqing has the largest emission reduction scale, with 10.19 μg/m^3^.

By multiplying the CO_2_ emissions, NO_X_ emissions and PM_2.5_ concentration that can be reduced in each province by the corresponding unit governance cost, the total atmospheric environment governance cost of the Yangtze River Economic Belt in 2025 can be obtained, as shown in Table 5. The total atmospheric environment governance cost in the Yangtze River Economic Belt will be CNY 3856.666 billion in 2025, accounting for 6.1% of the GDP of the entire economic belt. Anhui has the highest emission reduction cost at CNY 914.526 billion, accounting for 17.26% of this province’s GDP and 23.71% of the total emission reduction cost. Its NO_X_ emission reduction cost is CNY 450.694 billion, accounting for 49.28% of the province’s total emission reduction cost. The second is Hubei, the emission reduction cost of which is CNY 665.724 billion, accounting for 11.18% of this province’s GDP and 11.08% of the total emission reduction cost. The emission reduction costs of Shanghai, Jiangsu, Zhejiang and Sichuan are 0, and the emission reduction cost of each of the other provinces accounts for a high proportion of the GDP of the region, which will have a great impact on economic development.

### 3.2. Allocation of Atmospheric Environment Governance Cost in the Yangtze River Economic Belt

#### 3.2.1. Equitable Allocation of Atmospheric Environment Governance Cost

The number of possible alliances among 11 provinces in the Yangtze River Economic Belt is $2^{11}-1=2047$. Taking Shanghai as an example, the input–output data of provinces in alliance *S*, including Shanghai, are brought into the sequential SBM-Undesirable model to calculate the environmental efficiency values of each province under this alliance. Table 6 shows the environmental efficiency value of each province in the alliance that is composed of three provinces, including Shanghai, that is, the characteristic function value. At the same time, it shows the characteristic function value v(S) of the alliance *S* and that of the new alliance after excluding Shanghai, namely v(S/i).

According to the data in Table 6, when |S|=3, the value of $w\left|S\right|\frac{\left[v\left(S\right)/v\left(S/i\right)\right]}{v_{i}\left(S\right)}$ is 0.120. Similarly, the modified Shapley value of each province in the Yangtze River Economic Belt can be obtained as shown in Table 7.

It can be seen from Table 8 that the values of $\varphi_{i}\left(v\right)$ of Guizhou and Yunnan rank as the top two at 2.629 and 2.128, respectively, whereas the value of $\varphi_{i}\left(v\right)$ of Shanghai is the lowest at only 0.999.

According to Formula (12), the contribution rate of each province to the atmospheric environment governance of the Yangtze River Economic Belt is calculated. Then, the allocation scheme of atmospheric environment governance cost under the principle of fairness is obtained, as shown in Table 8.

The cost allocated to Guizhou and Yunnan account for a relatively large proportion, 0.1505 and 0.1216, respectively. The reason is that the environmental efficiency of these two provinces is low, and their emission reduction potential is far higher than that of other provinces, which indicates that these two provinces will need to increase emission reduction efforts in the future to improve the efficiency of atmospheric environment governance. Therefore, from the perspective of fairness, the two provinces should share more emission reduction costs. As provinces with relatively high environmental efficiency, Shanghai, Jiangsu, Zhejiang and Sichuan have relatively low emission reduction costs. This is because these four provinces have high economic development levels, significant emission reduction effects, and small emission reduction potential, and they are at the frontier of production. However, in the initial calculation, the emission reduction costs of Shanghai, Jiangsu, Zhejiang and Sichuan are zero. However, due to the mobility and spatial spillover effects of atmospheric emissions, and considering the regional coordinated governance of the atmospheric environment and the fairness of cost sharing, these provinces still need to bear some responsibility for reducing emissions.

By comparing the proportion of emission reduction cost in the GDP of each province in Table 5 and Table 8, it is found that, except for Yunnan and Hunan, the proportion of the remaining five provinces (excluding the four provinces with the initial allocation cost of 0) has decreased correspondingly. This shows that cost allocation based on the contribution of each province to the region has eased the financial pressure of some provinces to a certain extent and ensured the fairness and feasibility of regional environmental governance.

#### 3.2.2. Allocation of Atmospheric Environment Governance Cost Based on Modified FCA-DEA Model

According to the FCA-DEA model (14), the maximum and minimum costs allocated to each province are obtained, as shown in Table 9.

The total atmospheric environment governance cost is between CNY 2323.435 billion and 5384.971 billion. From a practical perspective, in order to alleviate the pressure of emission reduction, each province hopes to bear less responsibility and emission reduction cost. Therefore, the cost shared by each province will tend to be near the lower limit of the cost, so the total cost shared is only CNY 2323.435 billion, which is far less than the total cost to be allocated, that is, CNY 3856.666 billion. The minimum shared cost of Chongqing, Guizhou and Yunnan is 0, and the upper limit of the shared cost is far less than other provinces, only CNY 2615.03, 618.817 and 219.843 billion, respectively. The emission reduction cost range of Shanghai, Jiangsu, Zhejiang and Sichuan is much higher than that of other provinces. From the perspective of efficiency, Chongqing, Guizhou and Yunnan are located in the west of the Yangtze River Economic Belt, with backward economic development, imperfect emission reduction infrastructure, relatively backward technology, low efficiency of environmental governance, and insufficient use of emission reduction funds, so their shared costs are low. Shanghai, Jiangsu, Zhejiang and Sichuan are four provinces with high environmental efficiency, developed economies, complete infrastructure and rich experience in environmental governance, so they can afford more emission reduction costs.

In order to ensure the fairness and efficiency of the allocation scheme, the above solution results are brought into the model (16) to render the final allocation results as similar as possible to the schemes under the fairness principle in Table 9. The final allocation scheme is shown in Table 10. Where $f_{j}^{*}=a_{j}^{*}-b_{j}^{*}+f_{j}^{0}$, and $f_{j}^{0}$ is the allocation cost of each province under the principle of fairness in Table 10.

It can be seen from Table 10 that in 2025, Jiangsu, Zhejiang and Sichuan will share a relatively large amount of governance costs, which will be CNY 8189.28, 515.254 and 396.82 billion, respectively, accounting for 21.24%, 13.36% and 10.29% of the total cost to be allocated. Guizhou, Chongqing, Yunnan and Jiangxi will share less governance costs, which are CNY 1525.55, 2041.53, 2198.43 and 219.86 billion, respectively. It can be seen that although the modified FCA-DEA model has considered the principle of fairness and introduced fairness constraints, it more reflects the efficiency of allocation. Developed provinces such as Jiangsu, Zhejiang and Sichuan that can obtain higher output under the same input receive more governance costs. On the contrary, underdeveloped provinces such as Guizhou, Chongqing, Yunnan and Jiangxi that can obtain less output under the same input receive fewer governance costs.

### 3.3. Fairness Test of Allocation Scheme Based on the Modified FCA-DEA Model

The Gini coefficient of per capita cost can be used to compare the fairness of different cost allocation schemes. Gini coefficient formula is as follows [51]:(18)$$G=1-\sum_{i=1}^{11}\left(X_{i}-X_{i-1}\right)\left(Y_{i}-Y_{i-1}\right)$$
where $X_{i}$ and $Y_{i}$ are the proportion of the population and the proportion of the per capita cost accumulated to the *i*th province after ranking according to the per capita cost from high to low. The Gini coefficient of the cost allocation scheme based on the modified Shapley value is 0.2854, which is between 0.2 and 0.29, indicating that the allocation scheme is relatively average. The Gini coefficient of the cost allocation scheme based on the modified FCAM-DEA is 0.1933, which is less than 0.2, indicating that the allocation scheme is highly average. The smaller the Gini coefficient, the more equitable the allocation. Both allocation schemes achieve fairness of allocation, but the cost allocation scheme based on the modified Shapley value only defines the emission reduction responsibility of each province from the perspective of fairness and does not consider the use efficiency of funds. The allocation result based on the modified FCA-DEA model not only reflects the efficiency of the allocation but also further ensures the fairness of the allocation. Therefore, the allocation scheme based on the modified FCA-DEA model proposed in this paper is more reasonable.

## 4. Conclusions

This paper calculates the shadow prices of CO_2_, NO_X_ and PM_2.5_ in each province of the Yangtze River Economic Belt from 2013 to 2025. Based on the calculation of the emission reduction potential, the total atmospheric environment governance cost in the Yangtze River Economic Belt in 2025 is obtained. Combining the modified Shapley value model and FCA-DEA model, this paper proposes a modified FCA-DEA model and allocates the total atmospheric environment governance cost in 2025. The main conclusions are as follows:(1)From 2013 to 2025, the shadow prices of CO_2_, NO_X_ and PM_2.5_ emissions in each province of the Yangtze River Economic Belt show an upward trend, indicating increasing pressure for future emission reduction. The average shadow prices of CO_2_, NO_X_ and PM_2.5_ emissions in the Yangtze River Economic Belt are CNY 1432.65/ton, CNY 146.50 ten thousand/ton and CNY 3529.76 billion/(μg/m^3^), respectively. Zhejiang and Guizhou have the highest and lowest average shadow price of CO_2_ emissions, with CNY 2596.47/ton and CNY 748.63/ton, respectively. Shanghai and Sichuan have the highest and lowest average shadow price of NO_X_ emissions, with CNY 313.64 ten thousand/tonand 27.97 ten thousand/ton, respectively. Sichuan and Guizhou have the highest and lowest average shadow price of PM_2.5_ emissions, with CNY 17,296.39.(2)The average environmental efficiency of 11 provinces in 2025 is 0.68. The environmental efficiency values of Shanghai, Jiangsu, Zhejiang and Sichuan are all 1, and these provinces have no potential to reduce emissions. Guizhou has the largest CO_2_ and NO_X_ emission reduction potential, with 63.12% and 70.57%, respectively. Chongqing has the largest PM_2.5_ emission reduction potential of 34.31%. The total atmospheric environment governance cost in the Yangtze River Economic Belt will be CNY 3856.666 billion in 2025, accounting for 6.1% of GDP of the entire economic belt. Among the 11 provinces, Anhui has the highest emission reduction cost, accounting for 17.26% of the province’s GDP.(3)Based on the modified FCAM-DEA model, Jiangsu, Zhejiang and Sichuan will share a relatively large amount of governance costs in 2025, accounting for 21.24%, 13.36% and 10.29% of the total governance cost, respectively. Guizhou, Chongqing, Yunnan and Jiangxi will share fewer governance costs. The Gini coefficient of this cost allocation scheme is only 0.1933. The allocation scheme under the modified FCAM-DEA model achieves fairness as well as efficiency.

The results of this study provide useful inspiration for the formulation of regional atmospheric environment governance policies, and the proposed methods are also applicable in solving the problem of coordinated atmospheric environment governance in other regions. One limitation of this study is that only CO_2_, NO_X_ and PM_2.5_ are selected as atmospheric environmental emissions. In the future, the emissions of other greenhouse gases (such as CH_4_ and O_3_) and other air pollutants (such as CO and hydrocarbons) can be considered to be included in the model, so as to render the calculation and allocation of governance costs more in-depth.

## Figures and Tables

**Figure 1 ijerph-20-04281-f001:**
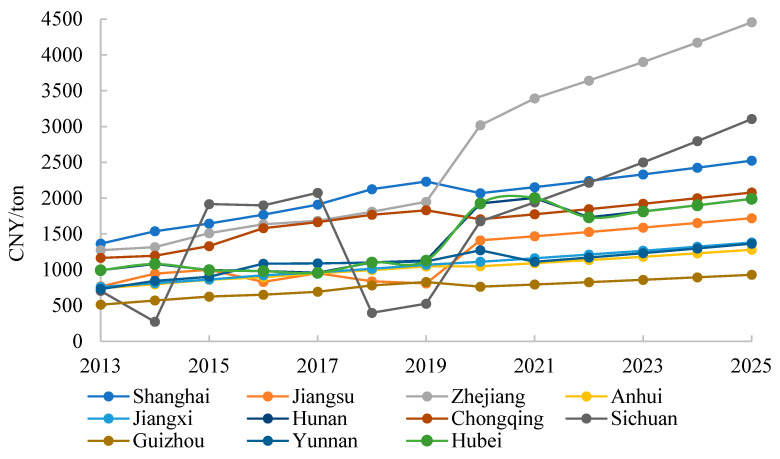
Shadow price of CO_2_ emissions in the Yangtze River Economic Belt from 2013 to 2025.

**Figure 2 ijerph-20-04281-f002:**
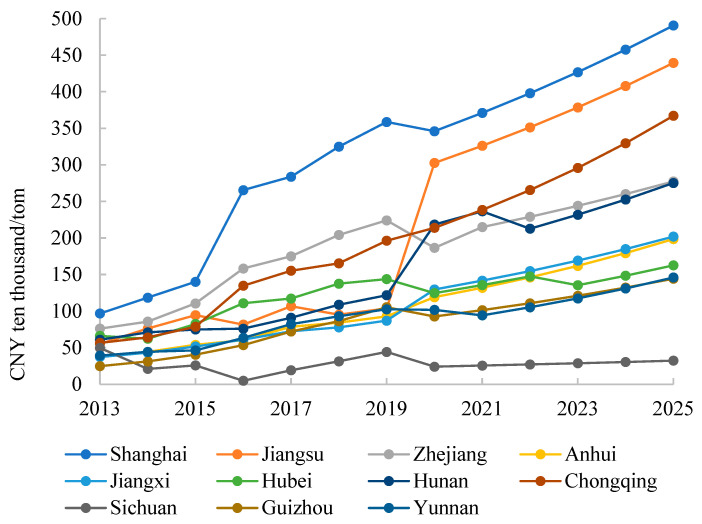
Shadow price of NO_X_ emission in the Yangtze River Economic Belt from 2013 to 2025.

**Figure 3 ijerph-20-04281-f003:**
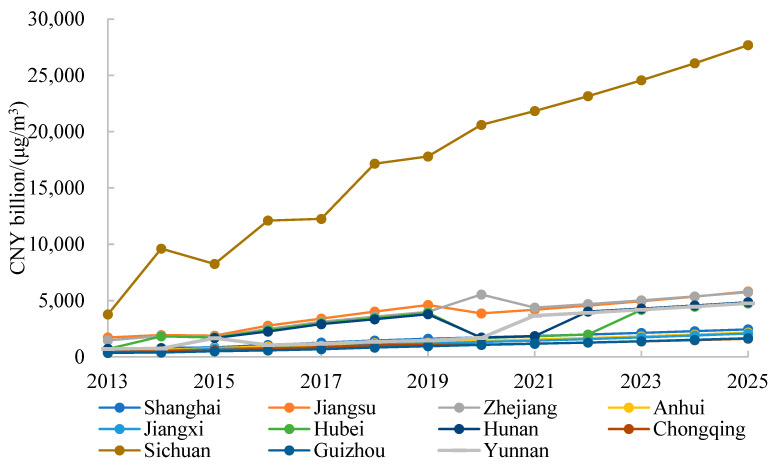
Shadow prices of PM_2.5_ in the Yangtze River Economic Belt from 2013 to 2025.

**Table 1 ijerph-20-04281-t001:** Descriptive statistics of variables from 2021 to 2025.

Indicator	Unit	Mean	Min	Max	St. Dev
Population	10^4^	5621.33	2502.86	9009.47	1947.01
Capital stock	CNY 10^8^	131,793.10	55,954.63	324,107.90	64,029.82
Energy consumption	10^4^ tce	17,937.82	9138.08	36,509.90	7455.561
GDP	CNY 10^8^	51,279.52	19,074.42	134,249.60	27,099.53
CO_2_ emissions	10^4^ tons	45,353.00	21,339.71	111,499.60	22,757.12
NO_x_ emissions	10^4^ tons	31.52	13.02	48.76	10.76
PM_2.5_ concentration	µg/m^3^	28.00	15.62	38.19	6.66

**Table 2 ijerph-20-04281-t002:** Forecast values of input and output indicators in 2025.

Province	Input Indicator			Desirable Output Indicator	Undesirable Output Indicator		
	Population (10^4^)	Capital Stock (CNY 10^8^)	Energy Consumption (10^4^ Tce)	GDP (CNY 10^8^)	CO_2_ (10^4^ Tons)	NO_X_ (10^4^ Tons)	PM_2.5_ (µg/m^3^)
Shanghai	2563	95,548.46	12,253.76	49,392.84	27,964.72	14.382	28.80
Jiangsu	9009	323,326.3	36,509.9	134,249.6	111,499.6	43.65	33.00
Zhejiang	7553	195,017.3	27,878.63	84,446.61	52,053.05	34.857	22.50
Anhui	6066	119,380.2	17,418.85	52,995.77	59,183.22	38.13	35.10
Jiangxi	4472	82,843.5	11,762.29	36,033.66	37,299.52	25.497	24.80
Hubei	5640	177,742.4	19,259.47	59,521.31	49,964.75	44.82	33.30
Hunan	6510	164,886.3	20,079.43	55,913.06	34,017.69	24.597	31.50
Chongqing	3413	85,696.74	10,272.86	33,459.37	22,985.98	13.02	29.70
Sichuan	8541	153,974.4	24,524.06	65,036.16	52,304.85	34.5	15.62
Guizhou	4216	72,299.81	12,886.01	25,002.67	38,379.4	24.741	21.52
Yunnan	4702	126,246.2	16,499.64	36,030.72	33,177.54	30.996	18.63

**Table 3 ijerph-20-04281-t003:** Comparison of average CO_2_ emission shadow price and average emission intensity.

Province	Shadow Price (CNY/Ton)	Emission Intensity (CNY Tons/10^4^)
Shanghai	2024.64	0.729
Jiangsu	1192.92	1.114
Zhejiang	2596.46	0.792
Anhui	1019.24	1.438
Jiangxi	1065.75	1.385
Hubei	1203.66	1.081
Hunan	1431.21	0.964
Chongqing	1681.33	0.878
Sichuan	1694.14	1.024
Guizhou	748.63	1.967
Yunnan	1101.18	1.257
Average value	1432.65	1.148

**Table 4 ijerph-20-04281-t004:** Environmental efficiency, output redundancy and emission reduction potential of provinces in the Yangtze River Economic Belt in 2025.

Province or City	Environmental Efficiency	CO_2_		NO_X_		PM_2.5_	
		Output Redundancy (10^4^ Tons)	Emission Reduction Potential (%)	Output Redundancy (10^4^ Tons)	Emission Reduction Potential (%)	Output Redundancy (μg/m^3^)	Emission Reduction Potential (%)
Shanghai	1.000	0.00	0.00	0.00	0.00	0.00	0.00
Jiangsu	1.000	0.00	0.00	0.00	0.00	0.00	0.00
Zhejiang	1.000	0.00	0.00	0.00	0.00	0.00	0.00
Anhui	0.529	29,178.62	49.30	22.70	59.53	4.20	11.96
Jiangxi	0.518	16,898.36	45.30	15.00	58.85	3.79	15.28
Hubei	0.540	15,163.73	30.35	27.35	61.02	0.00	0.00
Hunan	0.552	1497.76	4.40	8.21	33.36	0.00	0.00
Chongqing	0.579	4042.30	17.59	3.28	25.17	10.19	34.31
Sichuan	1.000	0.00	0.00	0.00	0.00	0.00	0.00
Guizhou	0.344	24,223.65	63.12	17.46	70.57	6.94	32.25
Yunnan	0.415	10,913.41	32.89	20.27	65.38	0.00	0.00
Average Value	0.680	9265.26	22.09	10.39	33.99	2.28	8.53

**Table 5 ijerph-20-04281-t005:** Atmospheric environment governance cost of the Yangtze River Economic Belt in 2025.

Province	CO_2_ (CNY 10^8^)	NO_X_ (CNY 10^8^)	PM_2.5_ (CNY 10^8^)	Total Cost (CNY 10^8^)	Proportion in GDP (%)
Shanghai	0.00	0.00	0.00	0.00	0.00
Jiangsu	0.00	0.00	0.00	0.00	0.00
Zhejiang	0.00	0.00	0.00	0.00	0.00
Anhui	3732.58	4506.94	905.74	9145.26	17.26
Jiangxi	2332.12	3029.38	786.57	6148.07	17.06
Hubei	2211.36	4445.89	0.00	6657.24	11.18
Hunan	298.03	2258.12	0.00	2556.15	4.57
Chongqing	840.59	1203.22	1640.06	3683.86	11.01
Sichuan	0.00	0.00	0.00	0.00	0.00
Guizhou	2254.39	2520.79	1151.93	5927.11	23.71
Yunnan	1489.15	2959.82	0.00	4448.97	12.35
Total cost	13,158.22	20,924.14	4484.30	38,566.66	6.10

**Table 6 ijerph-20-04281-t006:** Environmental efficiency values of provinces in the alliance.

Alliance	SH	JS	ZJ	AH	JX	HB	HN	CQ	SC	GZ	YN	v(S)	v(S/i)
(SH, JS, ZJ)	1	1	1	—	—	—	—	—	—	—	—	3.00	2.00
(SH, JS, AH)	1	1	—	0.53	—	—	—	—	—	—	—	2.53	2.00
(SH, JS, JX)	1	1	—	—	0.52	—	—	—	—	—	—	2.52	2.00
(SH, JS, HB)	1	1	—	—	—	0.54	—	—	—	—	—	2.54	1.61
(SH, JS, HN)	1	1	—	—	—	—	0.56	—	—	—	—	2.56	2.00
(SH, JS, CQ)	1	1	—	—	—	—	—	0.58	—	—	—	2.58	2.00
(SH, JS, SC)	1	1	—	—	—	—	—	—	1	—	—	3.00	2.00
(SH, JS, GZ)	1	1	—	—	—	—	—	—	—	0.34	—	2.34	1.40
(SH, JS, YN)	1	1	—	—	—	—	—	—	—	—	0.42	2.42	1.45
(SH, ZJ, AH)	1	—	1	0.53	—	—	—	—	—	—	—	2.53	2.00
(SH, ZJ, JX)	1	—	1	—	0.52	—	—	—	—	—	—	2.52	2.00
(SH, ZJ, HB)	1	—	1	—	—	0.55	—	—	—	—	—	2.55	2.00
(SH, ZJ, HN)	1	—	1	—	—	—	0.56	—	—	—	—	2.56	2.00
(SH, ZJ, CQ)	1	—	1	—	—	—	—	0.58	—	—	—	2.58	2.00
(SH, ZJ, SC)	1	—	1	—	—	—	—	—	1	—	—	3.00	2.00
(SH, ZJ, GZ)	1	—	1	—	—	—	—	—	—	0.34	—	2.34	1.45
(SH, ZJ, YN)	1	—	1	—	—	—	—	—	—	—	0.43	2.43	1.52
(SH, AH, JX)	1	—	—	0.53	0.52	—	—	—	—	—	—	2.05	2.00
(SH, AH, HB)	1	—	—	0.53	—	1	—	—	—	—	—	2.53	2.00
(SH, AH, HN)	1	—	—	0.53	—	—	1	—	—	—	—	2.53	2.00
(SH, AH, CQ)	1	—	—	0.53	—	—	—	0.58	—	—	—	2.11	2.00
(SH, AH, SC)	1	—	—	0.53	—	—	—	—	1	—	—	2.53	2.00
(SH, AH, GZ)	1	—	—	0.53	—	—	—	—	—	0.34	—	1.87	1.58
(SH, AH, YN)	1	—	—	0.53	—	—	—	—	—	—	1	2.53	2.00
(SH, JX, HB)	1	—	—	—	0.52	1	—	—	—	—	—	2.52	2.00
(SH, JX, HN)	1	—	—	—	0.52	—	1	—	—	—	—	2.52	2.00
(SH, JX, CQ)	1	—	—	—	0.52	—	—	0.58	—	—	—	2.10	2.00
(SH, JX, SC)	1	—	—	—	0.52	—	—	—	1	—	—	2.52	2.00
(SH, JX, GZ)	1	—	—	—	0.52	—	—	—	—	0.34	—	1.86	1.60
(SH, JX, YN)	1	—	—	—	0.52	—	—	—	—	—	1	2.52	2.00
(SH, HB, HN)	1	—	—	—	—	1	1	—	—	—	—	3.00	2.00
(SH, HB, CQ)	1	—	—	—	—	1	—	0.58	—	—	—	2.58	2.00
(SH, HB, SC)	1	—	—	—	—	0.56	—	—	1	—	—	2.56	2.00
(SH, HB, GZ)	1	—	—	—	—	1	—	—	—	0.34	—	2.34	1.60
(SH, HB, YN)	1	—	—	—	—	0.79	—	—	—	—	1	2.79	2.00
(SH, HN, CQ)	1	—	—	—	—	—	1	0.58	—	—	—	2.58	2.00
(SH, HN, SC)	1	—	—	—	—	—	0.57	—	1	—	—	2.57	2.00
(SH, HN, GZ)	1	—	—	—	—	—	1	—	—	0.34	—	2.34	1.58
(SH, HN, YN)	1	—	—	—	—	—	1	—	—	—	1	3.00	2.00
(SH, CQ, SC)	1	—	—	—	—	—	—	0.58	1	—	—	2.58	2.00
(SH, CQ, GZ)	1	—	—	—	—	—	—	0.58	—	0.34	—	1.92	2.00
(SH, CQ, YN)	1	—	—	—	—	—	—	0.58	—	—	1	2.58	2.00
(SH, SC, GZ)	1	—	—	—	—	—	—	—	1	0.34	—	2.34	1.55
(SH, SC, YN)	1	—	—	—	—	—	—	—	1	—	0.46	2.46	2.00
(SH, GZ, YN)	1	—	—	—	—	—	—	—	—	0.34	1	2.34	2.00

Note: SH is Shanghai; JS is Jiangsu; ZJ is Zhejiang; AH is Anhui; JX is Jiangxi; HB is Hubei; HN is Hunan; CQ is Chongqing; SC is Sichuan; GZ is Guizhou; YN is Yunnan.

**Table 7 ijerph-20-04281-t007:** The modified Shapley values of provinces in the Yangtze River Economic Belt.

Province	|S|										$\varphi_{i}\left(v\right)$
	2	3	4	5	6	7	8	9	10	11	
Shanghai	0.163	0.120	0.104	0.096	0.090	0.087	0.086	0.085	0.084	0.083	0.999
Jiangsu	0.168	0.128	0.115	0.109	0.106	0.105	0.105	0.105	0.105	0.105	1.151
Zhejiang	0.172	0.133	0.119	0.112	0.109	0.107	0.107	0.106	0.106	0.105	1.177
Anhui	0.186	0.155	0.149	0.150	0.154	0.159	0.164	0.171	0.177	0.185	1.650
Jiangxi	0.187	0.157	0.154	0.157	0.163	0.169	0.175	0.180	0.185	0.188	1.717
Hubei	0.184	0.156	0.155	0.160	0.166	0.172	0.176	0.179	0.180	0.181	1.709
Hunan	0.178	0.144	0.138	0.140	0.145	0.152	0.158	0.165	0.171	0.178	1.569
Chongqing	0.188	0.153	0.145	0.144	0.146	0.150	0.154	0.159	0.165	0.170	1.575
Sichuan	0.178	0.136	0.122	0.115	0.111	0.109	0.108	0.107	0.106	0.105	1.198
Guizhou	0.257	0.248	0.250	0.255	0.260	0.265	0.269	0.273	0.275	0.277	2.629
Yunnan	0.201	0.185	0.192	0.202	0.212	0.220	0.225	0.229	0.230	0.232	2.128

**Table 8 ijerph-20-04281-t008:** Allocation of atmospheric environment governance cost under the principle of fairness.

Province	Contribution Rate	Allocated Cost (CNY 10^8^)	Proportion in GDP (%)
Shanghai	0.0571	2201.83	4.46
Jiangsu	0.0658	2535.90	1.89
Zhejiang	0.0673	2593.76	3.07
Anhui	0.0943	3636.03	6.86
Jiangxi	0.0981	3783.26	10.50
Hubei	0.0977	3766.68	6.33
Hunan	0.0897	3457.67	6.18
Chongqing	0.0900	3469.74	10.37
Sichuan	0.0684	2638.99	4.06
Guizhou	0.1502	5793.95	23.17
Yunnan	0.1216	4688.85	13.01
Total	1	38,566.66	6.10

**Table 9 ijerph-20-04281-t009:** The maximum and minimum cost allocated to each province in the Yangtze River Economic Belt in 2025 based on FCA-DEA model.

Province	Minimum Cost ($L_{j}$) (CNY 10^8^)	Maximum Cost ($U_{j}$) (CNY 10^8^)
Shanghai	2176.20	5686.15
Jiangsu	8079.90	12,705.08
Zhejiang	5152.54	7822.93
Anhui	1636.56	4129.22
Jiangxi	978.50	2690.08
Hubei	1455.05	3862.82
Hunan	1624.39	4426.27
Chongqing	0.00	2615.03
Sichuan	2131.21	6188.17
Guizhou	0.00	1525.55
Yunnan	0.00	2198.43
Total	23,234.35	53,849.71

**Table 10 ijerph-20-04281-t010:** Allocation scheme for atmospheric environment governance cost of Yangtze River Economic Belt in 2025.

Province	$a_{j}^{*}$ (CNY 10^8^)	$b_{j}^{*}$ (CNY 10^8^)	$f_{j}^{*}$ (CNY 10^8^)
Shanghai	811.89	0.00	3013.72
Jiangsu	5655.38	0.00	8191.28
Zhejiang	2558.77	0.00	5152.54
Anhui	0.00	402.48	3233.55
Jiangxi	0.00	1584.65	2198.60
Hubei	0.00	134.97	3631.71
Hunan	0.00	46.11	3411.55
Chongqing	0.00	1428.21	2041.53
Sichuan	1329.21	0.00	3968.20
Guizhou	0.00	4268.40	1525.55
Yunnan	0.00	2490.43	2198.43

## Data Availability

The data used to support the findings of this study are available from the corresponding author upon request.

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
