# Peer review of "Calculation and Allocation of Atmospheric Environment Governance Cost in the Yangtze River Economic Belt of China"

_ijerph, 2023, doi:10.3390/ijerph20054281_

Round 1
Reviewer 1 Report
I really like your scientific research. It is modern and novel. The research purpose of your article is clearly stated, an appropriate method of study is applied and the limitations of study are explained. The conclusion part is well- written. Also, prospect for further research is provided.
Author Response
Dear Reviewer,
Thank you very much for your recognition of my work!
In addition, we improved the language in the text. At the same time, we made corresponding modifications according to the comments of other reviewers. Please see the contents marked in red in the text.
Thank you again!
Reviewer 2 Report
The paper presents the importance of calculating atmospheric environment governance cost and presents a model to obtain it. The paper is well-written and interesting also for other academic institutions and governments. There are several comments that the authors could address.
1. Line 417, chapter (2) Data from 2021 to 2025: Current situation in the world is rapidly changing; however, your model is based on past data. This is understandable; however, it would be beneficial if you would also consider the effects of the new reality. For example – did COVID-19 pandemic affect energy consumption, greenhouse gas emissions or other variables? And if the current media news is correct, the population in China is decreasing (https://www.bbc.com/news/world-asia-china-64300190). Does that affect the model?
2. Line 530, Table 5. Environmental efficiency, output redundancy and emission reduction potential of provinces in the Yangtze River Economic Belt in 2025: Is it correct that provinces with Environmental efficiency 1 and their variables in the table have a value of 0.00? If that is correct, please explain that in the text above the table.
Reviewer 3 Report
Dear Authors,
you have put a significant effort in investigating the issue of Atmospheric Environment Governance Cost in the Yangtze River Economic Belt of China. The paper has interesting findings, but in need to be rewritten in line to the journal's Instruction and there are also a few other items which could be improved.

Round 2
Reviewer 3 Report
Dear Authors,
thank you for considering my remarks while revising the paper and wishing you success in your further work.
Regards,
Reviewer 3